# Isolation Procedure for CP *E. coli* from Caeca Samples under Review towards an Increased Sensitivity

**DOI:** 10.3390/microorganisms9051105

**Published:** 2021-05-20

**Authors:** Natalie Pauly, Yvonne Klaar, Tanja Skladnikiewicz-Ziemer, Katharina Juraschek, Mirjam Grobbel, Jens André Hammerl, Lukas Hemmers, Annemarie Käsbohrer, Stefan Schwarz, Diana Meemken, Bernd-Alois Tenhagen, Alexandra Irrgang

**Affiliations:** 1German Federal Institute for Risk Assessment (Bundesinstitut für Risikobewertung), D-10589 Berlin, Germany; natalie_pauly@hotmail.com (N.P.); Yvonne.Klaar@bfr.bund.de (Y.K.); Tanja.Skladnikiewicz-Ziemer@bfr.bund.de (T.S.-Z.); Katharina.juraschek@bfr.bund.de (K.J.); mirjam.grobbel@bfr.bund.de (M.G.); Jens-Andre.Hammerl@bfr.bund.de (J.A.H.); annemarie.kaesbohrer@bfr.bund.de (A.K.); bernd-alois.tenhagen@bfr.bund.de (B.-A.T.); 2State Office for Health and Social Affairs Berlin, D-10639 Berlin, Germany; lukas.hemmers@lageso.berline.de; 3Unit for Veterinary Public Health and Epidemiology, University of Veterinary Medicine, AT-1210 Vienna, Austria; 4Institute of Microbiology and Epizootics, Freie Universität Berlin, D-14163 Berlin, Germany; Stefan.Schwarz@fu-berlin.de; 5FAO Reference Centre for Antimicrobial Resistance, Freie Universität Berlin, D-14163 Berlin, Germany; 6Institute of Food Safety and Food Hygiene, Freie Universität Berlin, D-14163 Berlin, Germany; diana.meemken@fu-berlin.de

**Keywords:** Isolation, carbapenemase, CPE detection, selective media, specificity, sensitivity

## Abstract

Due to the increasing reports of carbapenemase-producing Enterobacteriaceae (CPE) from livestock in recent years, the European Reference Laboratory for Antimicrobial Resistances (EURL-AR) provided a protocol for their recovery from caecum and meat samples. This procedure exhibited limitations for the detection of CPE with low carbapenem MIC values. Therefore, it was modified by a second, selective enrichment in lysogeny broth with cefotaxime (CTX 1 mg/L) and with meropenem (MEM 0.125 mg/L) at 37 °C under microaerophilic conditions. By Real-time PCR, these enrichments are pre-screened for the most common carbapenemase genes. Another adaptation was the use of in-house prepared MacConkey agar with MEM and MEM+CTX instead of commercial selective agar. According to the EURL-method, we achieved 100% sensitivity and specificity using the in-house media instead of commercial agar, which decreased the sensitivity to ~75%. Comparing the method with and without the second enrichment, no substantial influence on sensitivity and specificity was detected. Nevertheless, this enrichment has simplified the CPE-isolation regarding the accompanying microbiota and the separation of putative colonies. In conclusion, the sensitivity of the method can be increased with slight modifications.

## 1. Introduction

The World Health Organization defines carbapenems as ‘High Priority Critically Important Antimicrobials’ for human medicine [1]. This is due to their role in the treatment of severe human infections with multidrug-resistant bacteria. Up to now, in European countries carbapenems are not licensed for use in veterinary medicine. Nevertheless, carbapenems may be used in specific circumstances in companion animals [2,3]. In recent years, an increasing number of carbapenem resistant enterobacteria, including carbapenemase-producers, have been reported [4]. Interestingly, the reports are not only limited to human medicine [4,5,6] as carbapenem-resistant bacteria (i.e., *E. coli*, *Salmonella* enterica subsp. enterica) have also been detected repeatedly in environmental or animal samples [4,5]. Besides various chromosomally encoded mechanisms (i.e., changes in membrane permeability, efflux pumps), resistance to carbapenems is often mediated by degrading enzymes called carbapenemases that can hydrolyze carbapenems and most other β-lactams [7]. Typically, the corresponding carbapenemase genes are located on mobile (integrative and conjugative) genetic elements (i.e., plasmids) [8,9]. They exhibit different mechanisms of action (i.e., serine-based (i.e., KPC, OXA) and zinc-catalysed carbapenemases (i.e., NDM, VIM)) [10,11]. Carbapenemases usually show a broad activity against most β-lactams and are typically associated with resistance determinants against other antimicrobial classes often leading to a drastic reduction of therapeutic options in case of infections [12,13]. In the European Union (EU), OXA, VIM, NDM, and KPC carbapenemases are most frequently detected in human clinical settings [5,14,15]. In 2012, the first carbapenemase-producing (CP) *E. coli* was detected in livestock [16], and voluntary monitoring of CP *E. coli* in the EU member states along with the obligatory monitoring on extended spectrum β-lactamase (ESBL)/AmpC β-lactamases producing *E. coli* (CID 2013/652/EU) [17] was introduced. In 2021, through new legislation, the specific monitoring for carbapenem resistant *E. coli* became mandatory (CID 2020/1729/EU). For selective CP *E. coli* detection, the isolation method provided by the European Reference Laboratory for Antimicrobial Resistances (EURL-AR) (https://www.eurl-ar.eu/protocols.aspx, accessed on December 2019) is recommended. The method is based on a 1:10 non-selective enrichment of the samples in buffered peptone water (BPW), which needs to be inoculated onto suitable selective agars (i.e., the commercial chromID^®^ CARBA agar). Several studies described the chromID^®^ CARBA agar as one of the most sensitive and specific chromogenic media for the detection of CP Enterobacteriaceae (CPE) in clinical samples in human medicine [18,19,20,21]. In Germany, this agar represents one of the most commonly used selective media for CPE monitoring purposes [22]. By using a highly sensitive molecular pre-screening of enrichment cultures (i.e., by real-time PCR), laborious microbiological investigations might be avoided, and further efforts can focus on the recovery of CPE from presumptive-positive samples.

Unfortunately, for some CPE detected in German livestock the EURL-AR detection method failed in recent years [23,24]. For example, the isolate 17-AB02384 was not detected within the monitoring of CP *E. coli* but was recovered in the monitoring on ESBL/AmpC β-lactamase producing *E. coli* from the same sample [24]. Thus, we identified the necessity to improve the isolation method for CP *E. coli* from livestock samples, especially from caecum content. Difficulties in the microbiological processing of caeca samples are the growth of an accompanying microbiota, the low CPE concentrations, and the considerably lower MIC values of CPE from animal origin in comparison to human isolates [22]. Here, we present comparative analyses on the performance of the EURL-method with a number of alterations potentially leading to an improvement of the CPE detection and recovery rate. The focus of this study was on the recovery of CP *E. coli* from pig caecum, as the majority of the recently described CP *E. coli* from German livestock originate from this matrix. Therefore, samples were spiked and blinded to calculate the sensitivity and specificity of the approaches, which included different selective agars. 

## 2. Materials and Methods

### 2.1. Pre-Studies

Initial experiments were carried out to determine the growth and the survival of CPE in feces. Therefore, the survival rate (colony forming units (cfu)/g pig feces) of seven different CPE (four VIM-1 producing *E. coli*, two VIM-1 producing *Salmonella* (*S*.) Infantis and one NDM-1 producing *S. Corvallis*) was investigated over a period of 10 days. For the alternative method to be tested, spiked samples were prepared. For this, the seven CPE were inoculated each in lysogeny broth (LB) supplemented with 1 mg/L cefotaxime (CTX) (LB + CTX) and were incubated at 37 °C for 16–18 h. After incubation, 10 µL of the suspension was transferred into LB and was additionally incubated at 37 °C with 180 rpm until an optical density (OD_600 nm_) of 0.5 was reached. The suspensions were centrifuged for 10 min at 4000 rpm and resuspended in 0.9 % saline solution (*w/v*). Thereafter, 5 mL of the solution (~10^8^ cfu/mL) was applied to 45 g feces and homogenized for 1 min in a BagMixer 400SW (Interscience, Wiesbaden, Germany). The prepared samples were stored at 4–6 °C. Every 24 h, 1 g of the spiked samples was diluted in 0.9% saline solution to 10^−6^. Aliquots of 100 µL of the 10^−5^ and the 10^−6^ dilutions were plated out onto selective MacConkey agar (McC) supplemented with 0.125 mg/L meropenem (MEM) (McC+MEM) and 1 mg/L CTX and MEM (McC + CTX + MEM), and the cfu were counted after incubation for 20–24 h at 37 °C. 

To extend the time until sample processing in the laboratory, a potential improvement of the CPE survival in fecal samples was tested by adding stabilizing substances. Trehalose [25,26], glycerol [27,28], and sodium chloride [29,30] were tested as potential additives for the improvement of the recovery rate of the seven CPE in pig feces. Previous studies reported on the protective characteristics or the enhancive influence of these substances for the recovery and growth of Enterobacteriaceae [25,26,27,28,29,30]. Therefore, the described previous experiments were repeated by adding 1 mL of the supplement (trehalose, glycerol, and sodium acid) to 9 g of the caeca sample. The samples were thoroughly mixed for 1 min and a decimal dilution series up to 10⁻⁶ in 0.9% saline solution was prepared. Afterwards, 100 µL of the 10^−5^ and the 10^−6^ dilution were applied onto selective McC + CTX + MEM, incubated overnight at 37 °C, and cfu/g were determined.

The influence of different liquid media (Mossel medium [31] and LB) on CPE growth was tested to determine the most appropriate culture medium for the second enrichment step. Furthermore, possible influences of a sublethal concentration of MEM (0.008 µg/mL) [32] and the addition of ZnSO_4_ (70 μg/mL) [33] to the two liquid media were tested. Gullberg et al. (2014) [34] reported that sublethal antimicrobial concentrations are able to induce bacteria to retain plasmids and express corresponding genes.

Moreover, the growth of a CP *E. coli* and a CP *Salmonella* in LB + CTX and LB + MEM under aerobic and microaerobic conditions were compared. For spiking the samples, each CPE was inoculated in 4 mL BPW and incubated at 37 °C overnight. Thereafter, 50 µL of the culture was transferred to BPW and incubated at 37 °C and 180 rpm until an optical density (OD_600 nm_) of 1.0 was reached. From this suspension, 1 mL was added to 9 g fresh caeca sample. Thereafter, each 1 mL of this mix was transferred to 9 mL LB + CTX and incubated at 37 °C either under aerobic or under microaerobic conditions for 18 h. After incubation, 100 µL of the sample were plated four times onto McC + CTX + MEM and cfu were counted after 20–24 h of incubation at 37 °C. Potential differences induced by the two incubation conditions were assessed through a two-sided Wilcoxon signed rank test for independent samples.

### 2.2. Method Comparison

#### 2.2.1. Bacterial Strains

In total, twelve CP *E. coli* were used for the evaluation of both isolation methods and the comparison of the different selective media. These isolates cover a wide range of carbapenemase genes (*bla*_VIM-1_; *bla*_GES-5_; *bla*_KPC-2_; *bla*_NDM-1_ and *bla*_OXA-48_) (Appendix A). As reference for quality assurance concerning the selectivity for *E. coli* of the media, the *E. coli* ATCC 25922 was used. In all three approaches, six caeca samples were CPE-negative and inoculated with the *E. coli* ATCC 25922. In each of the other twelve caeca samples, one of the twelve CP *E. coli* was inoculated with a concentration of 100 cfu/mL each.

#### 2.2.2. Sample Preparation for the Method Comparison

All twelve CP *E. coli* and the *E. coli* control strain ATCC 25922 were inoculated in 4 mL BPW and incubated at 37 °C overnight. After incubation, 50 µL of the overnight culture was transferred to fresh BPW and incubated at 37 °C at 180 rpm until an optical density (OD_600_ nm) of 1 was reached. For each CP *E. coli*, 9 g fresh caeca sample was artificially contaminated with 1 mL of the 10^−5^ diluted bacterial suspension. For the control strain, six caeca samples were spiked. All spiked caeca samples were homogenized by thorough mixing for 1 min. After sample preparation, the samples were blinded by a non-participating person. All 18 samples were analyzed by two methods (Figure 1) as described below. First, 1 g of each spiked sample was added to 9 mL BPW and incubated at 37 ± 2 °C for 16–18 h. Thereafter, the samples were separately subjected to investigations using the two different isolation procedures. For reliable interpretation, the 18 blinded samples were investigated with both methods in three independent approaches.

#### 2.2.3. Isolation and Detection Methods

Method A represents the EURL-protocol for isolation of ESBL, AmpC and CP *E. coli* from caeca, (https://www.eurl-ar.eu/protocols.aspx, accessed in December 2019). For the method comparison, the commercial chromID^®^ CARBA SMART agar (bioMérieux, Nürtingen, Germany) was used. This agar is composed of the CARBA agar and the OXA agar in equal parts. Therefore, the EURL-protocol was adapted by applying 5 µL on a half agar plate instead of 10 µL on a full agar plate. This adaption allowed the agar validation as part of the method comparison (5 µL per half chromID^®^ CARBA SMART agar plate, 10 µL per a whole McC agar plate) (Figure 2). Additionally, a Real-Time PCR step was added after the enrichment in BPW (described below).

Method B represents the modified version of the isolation procedure. Therein, non-selective enrichment in BPW has been maintained, followed by a second selective enrichment step. For that purpose, 10 µL of the BPW pre-enrichment has been applied to 10 mL LB + CTX and 10 mL LB + MEM. Both enrichments were incubated at 37 ± 2 °C for 16–18 h under microaerophilic conditions. Thereafter, 5 µL of the LB + CTX enrichment was inoculated on the CARBA side, and 5 µL of the LB + MEM enrichment on the OXA side of the chromID^®^ CARBA SMART plate (Figure 1). In addition, 10 µL of LB + CTX or LB + MEM were inoculated on in-house selective agars McC + CTX + MEM or McC + MEM, respectively. Each sample was stroked on each agar in duplicate (six plates per sample). Plates were incubated for 16–18 h at 37 ± 2 °C. Subsequently, one colony with *E. coli* typical colony morphology was picked from each plate and enriched in 4 mL LB + MEM. The species of each colony was confirmed by PCR and MALDI-ToF. Confirmed colonies were incubated for 16–18 h at 37 ± 2 °C in LB and 500 µL aliquots were stored in 800 µL of 80% glycerol for further analysis. Samples spiked with OXA-producers were assessed as positive if growth was observed on the chromID^®^ OXA side, likewise on McC + MEM. For the other CP *E. coli*, the sample was regarded positive if growth was observed at least on chromID^®^ CARBA side or on McC + CTX + MEM. 

To confirm the validity of the results, the whole experiment was independently repeated three times. Moreover, to ensure that personal handling had no influence, both methods were carried out by two different people per approach. Moreover, each agar type was used in duplicate and deviations (i.e., colony morphology) were noted. If at least one colony could be isolated and confirmed from one of the duplicates, the sample was rated positive for the corresponding method and agar type.

#### 2.2.4. Typing

##### Antimicrobial Susceptibility-Testing (AST)

The antimicrobial susceptibility of the isolates (Appendix A) was determined by broth microdilution using defined antimicrobial substances and concentrations from the harmonized EU panel following Commission Implementing Decision (CID) 2013/652/EU [plates EUVSEC & EUVSEC2; TREK Diagnostic Systems (Thermo Fisher Scientific, Schwerte, Germany)]. AST was conducted according to the EN ISO20776-1:2006 [15] and MIC values were interpreted based on EUCAST definitions (www.eucast.org, November 2013) using epidemiological cut-off values fixed in CID 2013/652/2013. The strain ATCC 25922 was used for quality control. All isolates were tested twice. First, before using them for artificial contamination and second, after their successful recovery from the samples to assess possible changes in their antimicrobial resistance patterns.

##### MALDI-ToF MS

One putative colony of CP *E. coli* of each agar plate (six per sample) was chosen for species confirmation by MALDI-ToF MS. As matrix α-Cyano-4-hydroxycinnamic acid (HCCA, Bruker, USA) was used, and analyses were performed by MALDI Microflex Biotyper (Bruker Daltonics, Bremen, Germany) as recommended by the manufacturers. 

##### Molecular Typing of CP *E. coli*

Molecular detection of the carbapenemase genes was carried out to confirm the presence/absence of *bla*_GES_, *bla*_KPC_, *bla*_NDM_, *bla*_OXA-48_ and *bla*_VIM_ in enrichment cultures and in presumptive colonies. Both sample enrichments and presumptive colonies were used for extraction of template DNA by heat treatment of bacterial suspensions. For this extraction, 800 µL were centrifuged by 10,000× *g* for 5 min. The cell pellet was resuspended in 300 µL double distilled water and boiled for 10 min at 99 °C. Thereafter, the samples were cooled on ice and centrifuged for 2 min at 16,000× *g*. Finally, 255 µL of the supernatant was mixed with 45 µL trehalose for stable storage at −20 °C. Real-time PCR was performed on a Bio-Rad CFX system. Primers and probes were adapted from Swayne et al. (2011) [35] and van der Zee et al. (2014) [36]. Amplification was conducted using Biozym qPCR Mastermix (Biozym Scientific, Oldendorf, Germany) and the following conditions: Initially 2 min at 95 °C, 30 cycles with 5 s at 95 °C and 60 s at 60 °C for product amplification.

#### 2.2.5. Statistics

Results of both detection methods were expressed in percent (positive samples divided by all tested samples). The EURL-method and the modified method were compared in terms of their diagnostic accuracy indicated by sensitivity, specificity, false discovery and omission rates, and overall diagnostic accuracy. At this point, the blinding of the samples was dissolved. If the detected CP *E. coli* were confirmed using MALDI-ToF MS and real-time PCR, a sample was classified as positive. The sensitivity was expressed as the proportion of the positive samples correctly identified. The specificity was calculated as the proportion of the samples correctly identified as negative among all negative samples. False discovery rates were calculated as the proportion of all false positive samples among all positive samples. Likewise, false omission rates were calculated as the proportion of all false negative samples among all negative samples. Overall diagnostic accuracy was calculated as the proportion of all tests that gave a correct result. All estimates as well as their corresponding exact 95% confidence intervals (Clopper-Pearson 95% confidence intervals as precision measures honoring small sample sizes) were estimated using the epiR package in R [37].

## 3. Results

### 3.1. Modification of the EURL-Protocol for Isolation of CP E. coli from Caeca Samples

In order to determine the number of bacteria that survived the prevailing condition in the pig feces, the recovery rate of the bacteria in feces was determined after storage at 6 °C. Based on the cfu counts, a reduction of 2 log units within 24 h was detected in pig feces samples stored at 6 °C. The addition of glycerin, sodium chloride, and trehalose to the feces samples did not result in a positive effect on the CPE survival rate (Figure 3). 

In a first step, the suitability of two liquid media (LB- or Mossel-medium) as basis of this second enrichment were compared. The Mossel-medium was developed especially for Enterobacteriaceae [31], whereas LB-medium is a nutrient-rich microbial broth used for the cultivation of E. coli. No differences in the recovery rate were observed in the direct comparison of LB- or Mossel-medium as second enrichment (data not shown). As LB is a common medium used in many laboratories, it was used for further investigations.

The addition of sublethal conditions of ZnSO_4_ and MEM was intended to activate potentially present resistance mechanisms in the liquid culture so that the corresponding strains may have a growth advantage when subsequently plated onto the selective solid media. However, the slight selection pressure did not result in a better recovery rate. Therefore, we used the second enrichment in LB + CTX as previously described [23]. To isolate the OXA-48 producers as well, we tested another second enrichment in LB + MEM. Enterobacteriaceae are described as facultative anaerobes [38]. Therefore, the target species can survive in both aerobic and anaerobic environments, in contrast to some representatives of the accompanying microbiota. The microaerophilic incubation led to a median increase in isolated CP E. coli from 7 cfu/mL with aerobic incubation to 26.5 cfu/mL (Wilcoxon test, W = 0, *p* = < 0.05, *n* = 8). Likewise, the median number of CP Salmonella increased from 81 cfu/mL with aerobic incubation to 144.5 cfu/mL (Wilcoxon test, W = 0, *p* = < 0.05, *n* = 8).

Based on the above-mentioned observations, the procedure of the method was modified to promote the growth of CPE and to reduce the amount of accompanying microbiota (Figure 4). The modified procedure includes a first, non-selective enrichment in BPW, followed by two parallel selective enrichments (LB + CTX and LB + MEM) incubated under microaerophilic conditions. Aliquots (10 µL) of the second enrichment were applied onto McC agar supplemented either with MEM or CTX + MEM. In the following chapter, this modified method was compared to the EURL-method.

### 3.2. The Comparison of the Modified Method with the EURL-Method for Isolation of the CP E. coli from Caeca Samples

To determine the performance of the modified method (method B), its diagnostic accuracy was compared to the reference method of the EURL-AR, the official protocol for isolation of ESBL, AmpC and CP E. coli from caeca samples (method A). The complete procedure of the comparison and validation is schematically illustrated in Figure 1. In the following, the results of the three repetitions were considered together (sample size *n* = 54) for each method and agar type (Table 1). 

Comparison of diagnostic accuracy measures show that the use of method A in combination with the in-house agar led to a significantly higher sensitivity, relative to the use of method A with the chromID^®^ SMART CARBA agar. In addition to this overall evaluation, the results for the individual approaches are given in Appendix A. The best CPE detection was achieved by using the original EURL-protocol in combination with in-house prepared McC + MEM and McC + CTX + MEM agar plates. All negative samples (*n*/N) were correctly recognized, and CP E. coli were successfully recovered from each of the artificially contaminated samples (*n*/N). This corresponds to a 100% sensitivity and specificity. The sensitivity decreased to 75% when the in-house agar plates were replaced by the commercial chromID^®^ CARBA agar. However, specificity remained at 100%. The modified method (B) reached a sensitivity of 86.1% when using the in-house agar and a sensitivity of 66.7% when using the commercial agar. However, all negative samples (*n*/N) were also correctly identified leading to a specificity of 100%. The added value of this modified method is the simplification of the isolation of putative colonies that are difficult to isolate. This could be shown by counting the agar plates (108 per agar type and method) with and without accompanying microbiota (i.e., Pseudomonas (P.) putida, P. monteilli, P. aeruginosa, Proteus mirabilis) (Table 2). Using the EURL-procedure in combination with the chromID^®^ CARBA agar, on 44.5% (48/108) of the agar plates growth of accompanying microbiota was observed. That was reduced to 39.8% (43/108) using the modified method with the chromID^®^ CARBA agar. In addition, for the chromID^®^ OXA agar, a reduction of accompanying microbiota from 23.2% (25/108) of the plates by using the EURL-procedure to 15.7% (17/108) using the modified method was observed. Using the in-house agars and the EURL-procedure, accompanying microbiota was observed on 37.0% (40/108) of the McC + MEM plates and on 33.4% (36/108) of the McC + CTX + MEM plates. By changing to the modified method, we detected accompanying microbiota on 29.6% (32/108) of the McC + MEM plates and on 36.1% (39/108) of the McC + CTX + MEM plates.

### 3.3. Molecular Detection of the Carbapenemase Genes

As previously described, the multiplex real-time PCR was used to pre-screen the enrichments (BPW, LB + CTX, and LB + MEM). Discrepancies were detected between the PCR and the results of the culture-based isolation. Comparing the PCR results with the true status of the samples, a sensitivity of 61.1% and a specificity of 94.4% was calculated for the molecular screening of the heat-treated BPW enrichments for carbapenemase genes. Fourteen out of thirty-six positive samples were not correctly identified. One of the 18 negative samples was incorrectly identified as positive for the bla_GES_.

When considering the screening of both second enrichments (results listed in Table 3), just two of the false negatives remained false negative, and the false positive sample remained as well. Therefore, the PCR achieved a sensitivity and a specificity of 94.4% by using both second enrichments as DNA template. Moreover, the real-time PCR showed erroneous simultaneous detection of bla_NDM_ and bla_VIM_ in various samples (9.2%) that were only bla_VIM_-positive. However, a high cp-valuefor bla_NDM_ was detected, suggesting low amplification. The results for the weekly approaches are given in Appendix A.

## 4. Discussion

### 4.1. Modification Steps of the Official Isolation Protocol 

All modifications were tested to reduce the amount of accompanying microbiota or to increase the recovery rate of the CPE. First, the recovery rate of the target bacteria in pig feces was determined at 6 °C and based on the bacterial growth calculated by the development of the cfu/g every day. To increase the survival of bacteria for better isolation, several additives to the pig feces were tested. In previous studies, the addition of sodium chloride has been reported to support growth under difficult conditions like low pH and high lactate concentrations [29]. Another study evaluated trehalose for the protection of *E. coli* against carbon stress [25]. The absence of a positive effect of these additives in our study may be due to the lack of a clear definition of the individual influencing factors of each fecal sample, i.e., the microbiota and other factors, which could have an influence on the nature of the samples. It should be noted here that the detection of CPE was increasingly challenging over the investigated period. One major challenge was the differentiation of CPE from increasing amounts of accompanying flora. Therefore, we recommend a timely processing of samples, preferably within 24 h after collection as this was an important factor for reliable detection of CP *E. coli* in feces and food matrices. 

After estimating bacterial survival, other options for improved detection of the target bacteria were tested. To account for the low MIC values of CPE isolated from samples of animal origin [22], we considered triggering gene expression by using a sublethal concentration of MEM. Moreover, the usage of ZnSO_4_ to push the metallo-β-lactamases was tested. Both approaches did not improve the recovery rate. Previous experiments achieved a beneficial effect by using a second selective enrichment step. This enrichment had been conducted in LB+CTX [23]. As OXA-48 producers are not reliably able to survive the presence of other β-lactams, like CTX, we considered another parallel second enrichment using LB+MEM [39,40]. The usefulness of the combination of both enrichments was also proven by the sensitivity of 94.4% of the Real-Time PCR. 

Further, we exchanged the commercial selective agar chromID^®^ CARBA agar with an in-house prepared McC + CTX and McC + CTX + MEM. Our results support a previous study on the suitability of chromID^®^ CARBA agar for detecting CPE with only slightly reduced susceptibility to carbapenems [22]. Within this study, it is presumed that the concentration of different ingredients and thus the selectivity is too high for CPE originating from animals or food. CPE from livestock often exhibit lower MIC values for the respective carbapenems than clinical human isolates [22,24]. Our previous study investigated the use of chromID^®^ CARBA after the end of the shelf life expecting a possible decrease in antimicrobial concentration. As the selective agar is not specified in the EURL-AR reference method (https://www.eurl-ar.eu/protocols.aspx, December 2019), the change to adjusted agar might be easy adaptable. In this study, we analyzed the use of an in-house McC agar, supplemented with MEM and MEM+CTX, and commercial agar for the detection of CPE from feces. The experiments were carried out with fresh prepared in-house agars (two weeks). However, the comparison in this study is limited to only one commercial selective agar. For commercial agars, a wide variation regarding their suitability for the isolation of CPE from different samples has been reported [18,20,41,42]. For the chromID^®^ CARBA agar, sensitivity in the literature ranges from 30 to 100%, and specificity from 50 to 100%. Still, the chromID^®^ CARBA agar is described as one of the most sensitive media [18,20,41,42] and is therefore commonly used by German laboratories in the monitoring according to CID 2013/652/EU. 

### 4.2. The Comparison of Both Microbiological Methods 

Although the isolation of CPE from livestock and food is rare in monitoring programs, the number of reports on successful recovery is increasing in recent years [6]. The isolation methods described differ from direct plating to different incubation conditions, supplements, and enrichments [23,43,44]. These non-harmonized detection methods in research projects in the animal sector complicate a realistic assessment of the occurrence of CPE in livestock. The EURL-AR has developed and provided a harmonized method for CP *E. coli* isolation from meat and caeca samples, which is meant to be used in monitoring according to CID 2013/652/EU. However, it is currently rarely used beyond the EU monitoring programs. Several publications reported on difficulties in detecting CPE with the EURL-method [22,23,24]. We observed a sensitivity of 75% and specificity of 100% for the EURL-AR protocol with a commonly used commercial agar. This implies that 25% of CPE-positive samples were not recognized, which demonstrates the need for an improvement of the reference method. Our results indicate that using the validated EURL-protocol in combination with the non-commercial McC agar supplemented with 0.125 mg/L MEM, and with 0.125 mg/L MEM and 1 mg/L CTX significantly improved the sensitivity compared to all other approaches. Selective enrichment under microaerobic conditions did not achieve the desired effect. This microaerobic incubation was promising because of the ability of Enterobacteriaceae to grow under this condition, in contrast to e.g., Aeromonads, which are often part of the accompanying microbiota [38]. Other options to favor Enterobacteriaceae over other bacteria have been investigated. An incubation temperature of 44 °C was described in the report on the validation of selective McC agar supplemented with 1 mg/L CTX for monitoring of ESBL- and AmpC producing *E. coli* in meat and caeca samples [45] and in a previous study of Irrgang et al. 2019 [23]. The authors recognized that the growth of CP Enterobacteriaceae other than CP *E. coli* might also be inhibited [23]. In terms of harmonization, a method should be capable of also detecting other bacterial species, as all CPE are of interest regarding the distribution of the resistances [6]. Our study focused on *E. coli* as the EU monitoring for ESBL/pAmpC and carbapenemases targets this species.

The reported problems during detection of CPE-containing samples indicate a low occurrence of CPE in samples of animal origin [23,24]. Another rarely discussed reason for reduced detection of CPE from feces and caecum could be that the bacteria in this environment have lost the ability to grow on routine media. It is described that *E. coli*, *Klebsiella pneumoniae*, *Salmonella enterica* serovar Typhimurium, and other human/zoonotic pathogens could enter a distinct state, a so-called viable but non-culturable (VBNC) state [46,47]. In combination with the physical stress exerted by caeca samples, the transition to a VBNC status would be useful for the survival of CPE [47]. Overall, we need to be able to detect positive samples and to isolate these CPE for a further characterization and risk assessment. 

Another approach to improve sensitivity might be a more sensitive molecular screening method. Using pre-screening by PCR, efforts can focus on the cultivation of the isolates from PCR-positive samples. This would decrease the costs and the personal time necessary for the microbiological isolation. The multiplex Real-Time PCR applied in this study showed good results when screening the second enrichment (a sensitivity and specificity of 94.4%). It might be possible that the use of a quantitative Real-Time PCR will provide more reliable results as a direct increase of the bacterial growth can be detected and used for adapting the time points for selective cultivation. The Real-Time PCR performed was not effective when applied on the initial BPW enrichment. To avoid the isolation step of a target bacterium and to immediately analyze the entire sample, the option of metagenomics becomes more attractive. Metagenomics would allow a simultaneous identification and typing of CPE and other pathogens [47]. It would be a powerful tool for monitoring purposes and might be considered in the long term to become part of the routine method. Currently, this method is too expensive for routine use and requires additional bioinformatic expertise.

## 5. Conclusions

After various approaches to optimize the recovery rate of CPE in pig feces, it could be concluded that potential additives did not have a positive effect. However, a positive effect can be achieved by processing the samples in a timely manner. Moreover, our results confirmed the influence of the chosen media. Typically used commercial agar plates are optimized for the isolation of CPE from human clinical samples with usually high MIC values for carbapenems. Therefore, we recommend the EURL-method with a selective medium corresponding to the searched target bacterium. A second enrichment under microaerophilic conditions reduced the accompanying microbiota, but the sensitivity of the whole procedure was decreased from 100% to 86.1%. The PCR-screening from a second enrichment showed good results. Therefore, we recommend the modified method if a presumptive CP *E. coli* cannot be isolated. However, a sensitive method is essential to avoid an underestimation of the CPE occurrence in livestock in Europe. 

## Figures and Tables

**Figure 1 microorganisms-09-01105-f001:**
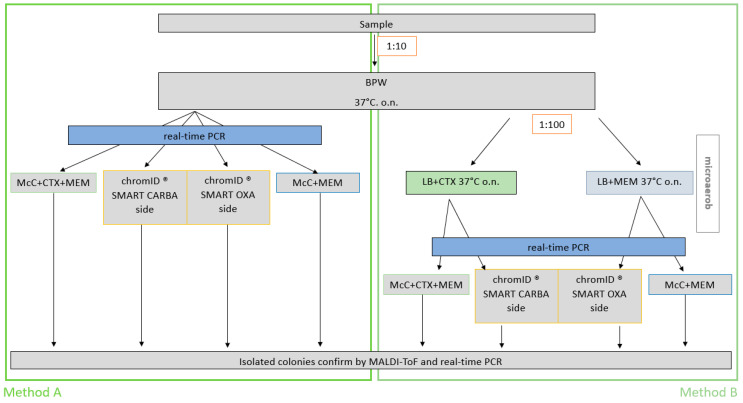
The whole procedure of method A and method B to compare both and to validate the agar plates. BPW = Buffered Peptone Water; McC + CTX + MEM = MacConkey Agar supplemented with 1 mg/L cefotaxim and 0.125 mg/L meropenem; McC + MEM = MacConkey Agar supplemented with 0.125 mg/L meropenem; LB + CTX = lysogeny broth supplemented with 1 mg/L cefotaxim; LB+MEM = lysogeny broth supplemented with 0.125 mg/L meropenem.

**Figure 2 microorganisms-09-01105-f002:**
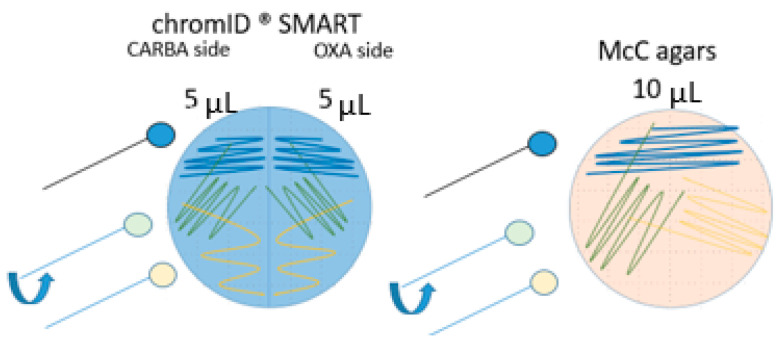
Spreading the sample on the plate. The first spread (blue, first third of the plate) was performed by using a 10 µL-loop. A second spread (green, second third of the plate) was performed by another 10 µL-loop, which was turned the last third of the spread (yellow).

**Figure 3 microorganisms-09-01105-f003:**
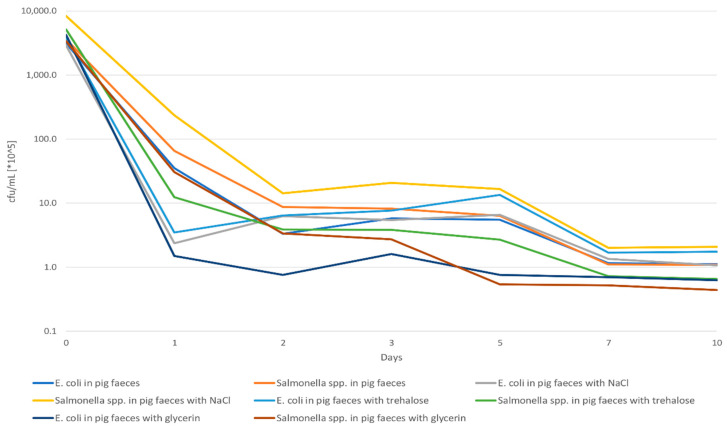
The decreasing number of CPE (*E. coli* and *Salmonella* each in average) in pig faeces with and without some additives over ten days.

**Figure 4 microorganisms-09-01105-f004:**
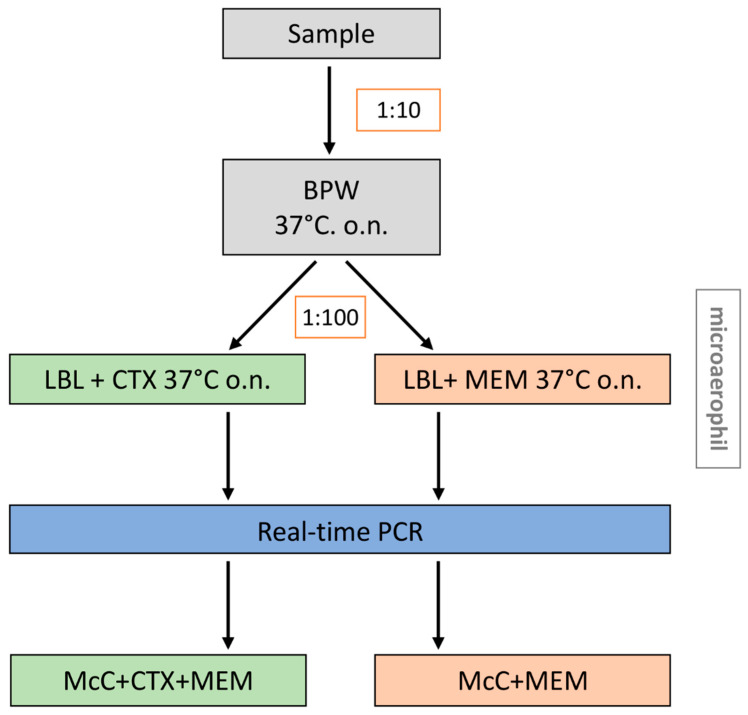
The final step-by-step guidance of the modified method (method B). BPW = Buffered Peptone Water; McC + CTX + MEM = MacConkey Agar supplemented with 1 mg/L cefotaxim and 0.125 mg/L meropenem; McC + MEM = MacConkey Agar supplemented with 0.125 mg/L meropenem; LB + CTX = lysogeny broth supplemented with 1 mg/L cefotaxim; LB + MEM = lysogeny broth supplemented with 0.125 mg/L meropenem.

**Table 1 microorganisms-09-01105-t001:** Summarized results on the repeated isolation of CPE from the spiked fecal samples (calculation by combining the OXA and CARBA- site of the ChromID^®^ SMART CARBA agars and of the McC + CTX + MEM and the McC + MEM plate of the in-house agar). Method B is the modified method. Corresponding exact 95% confidence intervals are provided in brackets.

	Method A + ChromID^®^ SMART CARBA agars	Method A + *in-house* agars	Method B + *in-house* agars	Method B + ChromID^®^ SMART CARBA agars
Sensitivity	75 (57.8–87.9)	100 (90.2–100)	86.1 (70.5–95.3)	66.7 (49–81.4)
Specificity	100 (81.5–100)	100 (81.5–100)	100 (81.5–100)	100 (81.5–100)
False discovery rate	0 (0–12.8)	0 (0–9.7)	0 (0–11.2)	0 (0–14.2)
False omission rate	33.3 (16.5–54)	0 (0–18.5)	21.7 (7.5–43.7)	40 (22.7–59.4)
Accuracy	83.3 (70.7–92.1)	100 (93.4–100)	90.7 (79.7–96.9)	77.8 (64.4–88)

**Table 2 microorganisms-09-01105-t002:** Validation of the selective agars by counting the number of plates with growing of accompanying microbiota for each group per used method. McC + CTX + MEM = MacConkey Agar supplemented with 1 mg/L cefotaxim and 0.125 mg/L meropenem; McC + MEM = MacConkey Agar supplemented with 0.125 mg/L meropenem.

	ChromID^®^ SMART CARBA Side	ChromID^®^ SMART OXA Side	McC + MEM	McC + CTX + MEM
Method A	48/108 (44.45%)	25/108 (23.15%)	40/108 (37.04%)	36/108 (33.36%)
Method B	43/108 (39.82%)	17/108 (15.74%)	32/108 (29.63%)	39/108 (36.14%)

**Table 3 microorganisms-09-01105-t003:** Accumulated results on the weekly detection of carbapenemase genes from the spiked fecal samples. BPW was used for the first enrichment, LB + CTX and LB + MEM were used parallel as second enrichments. Corresponding exact 95% confidence intervals are provided in brackets.

	BPW	LB + CTX	LB + MEM	Both LB Enrichments
Sensitivity	61.1 (43.5–76.9)	83.3 (67.2–93.6)	58.3 (40.8–74.4)	94.4 (81.3–99.3)
Specifity	94.4 (72.7–99.6)	94.4 (72.7–99.6)	100 (81.5–100)	94.4 (72.7–99.9)
False discovery rate	4.3 (0.1–21.9)	3.2 (0.1–16.7)	0 (0–16.1)	2.9 (0.1–14.9)
False omission rate	45.2 (27.3–64)	26.1 (10.2–48.4)	45.4 (28.1–63.6)	10.5 (1.3–33.1)
Accuracy	72.2 (58.4–83.5)	87.0 (75.1–94.7)	72.2 (58.4–83.5)	94.4 (84.6–98.8)

## Data Availability

MDPI: Research Data Policies.

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
