# Peer review of "Isolation Procedure for CP E. coli from Caeca Samples under Review towards an Increased Sensitivity"

_microorganisms, 2021, doi:10.3390/microorganisms9051105_

Round 1
Reviewer 1 Report
The authors compare different detection methods based on spiked samples. This gives them a gold-standard for positivity (those that were spiked).
I cannot judge the experimental set-up, but the statistical analyses miss essential steps. Namely the precision of the estimates is not indicated, such that point estimate of sensitivity or specificity does not mean much with small sample sizes.
The authors should include confidence intervals (or lower limits) for their estimates. This is simply done with any statistical package (e.g. function binom.test in R). The limit for the sensitivity of 100% for example is based on 12 samples, thus the lower confidence limit is 74% which is less than the point esimate for the other method! The difference in test characteristics is not tested and could be by coincidence. A difference of 9 out of 12 versus 12 out of 12 successes is likely to be a conincidental.
The positive predictive value and negative predicted values are calculated, but how is not explained in the M&M and again no confidence intervals are given. Also for the accuracy I need to guess that this is the Rand index (without confidence interval).
This manuscript would definitely benefit from quantifying the precision of the estimates. I doubt whether a statistical significant difference between the methods is present due to the small sample size. The statistics that need to be done are simple (I teached them to second year veterinary students) and should be standard for researchers to apply and could be done within a day.
Author Response
Dear Reviewer,
we appreciate your detailed and specific feedback regarding the statistical analysis in this manuscript. We implement the exact 95% confidence intervals in table 1 and 3. The previous table 3 (weekly calculation) is changed to Supplemental Material Table S2. We agree, that for the small sample number (n=18) a statistical calculation is not reasonable. The recommended calculations show that a significant improvement in sensitivity could be achieved for method A combined with in-house medium (line 300-303).
Moreover, we added a more detailled describtion of the statistical calculation for table 1 and 3 (line 232-245). In line 417-421, the calculated effect is discussed.
Accordingly we have added Clopper-Pearson 95% confidence intervals as precision measures honouring small sample sizes.
Kind regards,
Alexandra Irrgang
Reviewer 2 Report
A useful paper with very thorough method development. Generally well written, just a few points of concern
Abstract and throughout the manuscript - "suspicious" colonies is not proper usage. Would suggest 'putative' or 'suspect'
L28 delete 'been'
L38 delete 'just'
L47 carbapenemase genes also on ICE elements. See Botelho et al. 2018.
L210 suspect colony of CP E. coli?
L335 Late in coming? Needs further description. Slow amplification???
L343 'development of the number' - do you mean bacterial growth?
L408 what other study? More description is needed.
L411 would suggest 'capable of also detecting...'
Author Response
Dear Reviewer,
thank you for your suggestions. We implement the proposed improvements and the recommended publication (Botelho et al. 2018) in the manuscript and marked changes in the manuscript files using the "Track Changes" function in Microsoft Word. Following the exact changes:
L29: ´been´deleted
L39: ´just´deleted
L48: Sentence changed to ´Typically, the corresponding carbapenemase genes are located on mobile (integrative and conjugative) genetic elements (i.e. plasmids)´ and the reference was added.
L212: Sentence changed to ´One putative colony of CP E. coli of each agar plate…´
L352 and 353: Sentence changed to ´However, a high cp-value for blaNDM was detected, suggesting low amplification.
L355 and 356: Sentence changed to ´… based on the bacterial growth calculated by the development of the cfu/g every day
L429: The exact study “Irrgang et al., 2019” was added
L432: Changed to ´capable of also detecting´
Kind regards,
Alexandra Irrgang
Round 2
Reviewer 1 Report
The authors have provided the requested confidence intevals in an appropriate way.
One remark remains now that the calculation of the PPV and NPV are shown. In epidemiology both refer to the probability of being positive (or negative) given a positive (or negative) test in the study population. For clinical relevance this needs to be calculated with an estimate of the prevalence in a case-control study. This experimental study reflects a case-control setting. The cases are the spiked samples and the controls are evidently not. Therefore these numbers do not carry clinical relevance, because each epidemiologist knows that the PPV and NPV depend on the prevalence.
PPV increases with prevalence, because the fraction of false positives decreases. because less false positives are possible. NPV decreases with the prevalence because the fraction of true negatives decreases.
I suggest that the PPV and NPV are removed and potentially replaced by the false discovery rate and false ommission rate are used.
Author Response
Dear Reviewer,
according to your suggestions, we have replaced the PPV and NPV estimates with the proposed false discovery rate and false omission rate estimates and corresponding exact confidence intervals. Thank you for your thoughtful review and detailed explanations.
Kind regards,
Alexandra Irrgang